# A Study on Synthesis and Upscaling of 2′-*O*-AECM-5-methyl Pyrimidine Phosphoramidites for Oligonucleotide Synthesis

**DOI:** 10.3390/molecules26226927

**Published:** 2021-11-17

**Authors:** Kristina Karalė, Martin Bollmark, Rouven Stulz, Dmytro Honcharenko, Ulf Tedebark, Roger Strömberg

**Affiliations:** 1Department of Biosciences and Nutrition, Karolinska Institutet, Neo, 141 57 Huddinge, Sweden; kristina.druceikaite@ki.se (K.K.); rouven.stulz@ki.se (R.S.); dmytro.honcharenko@ki.se (D.H.); 2RISE, Department Chemical Process and Pharmaceutical Development, Forskargatan 18, 151 36 Södertälje, Sweden; martin.bollmark@ri.se (M.B.); ulf.tedebark@ri.se (U.T.); 3Oligonucleotide Discovery, Discovery Sciences, BioPharmaceuticals R&D, AstraZeneca, 431 50 Gothenburg, Sweden

**Keywords:** 2′-*O*-(N-(aminoethyl)carbamoyl)methyl modification, phase transfer catalysis (PTC), 5-methyluridine, 5-methylcytidine, oligonucleotides, alkylation, monoacetylation

## Abstract

2′-*O*-(*N*-(Aminoethyl)carbamoyl)methyl-modified 5-methyluridine (AECM-MeU) and 5-methylcytidine (AECM-MeC) phosphoramidites are reported for the first time and prepared in multigram quantities. The syntheses of AECM-MeU and AECM-MeC nucleosides are designed for larger scales (approx. 20 g up until phosphoramidite preparation steps) using low-cost reagents and minimizing chromatographic purifications. Several steps were screened for best conditions, focusing on the most crucial steps such as N^3^ and/or 2′-OH alkylations, which were improved for larger scale synthesis using phase transfer catalysis (PTC). Moreover, the need of chromatographic purifications was substantially reduced by employing one-pot synthesis and improved work-up strategies.

## 1. Introduction

In recent years, therapeutic oligonucleotides (ONs) have become one of the most promising classes of drug candidates for the treatment of life-threatening diseases [1]. Nevertheless, different therapeutic approaches, such as antisense oligonucleotides (ASOs) [1,2,3], RNA interference [4,5,6] and microRNA [1,7] technologies, usually face two types of limitations: oligonucleotide instability in biological fluids and problematic delivery to the site of action. Modifications of both the base and the sugar fragments of an oligonucleotide were shown to provide several advantages in the past [1]. For instance, introduction of a methyl group to the 5th position in cytosine enhances duplex thermal stability [1]. Methylation of pyrimidines also reduces the immune response [8,9,10,11]. 2′-*O*-Alkyl oligonucleotide modifications [12] showed several advantages like binding affinity and resistance to nucleases [13], including a relatively straightforward synthesis, allowing potentially cost-efficient manufacturing. The synthesis of 2′-aminoethoxy-modified oligonucleotides was shown to increase the binding affinity [14]. Later, using an alternative pathway, allowing introduction of different bases at a later stage into 2′-aminoethoxy-modified nucleoside monomers was introduced [15]. The 2′-*O*-carbamoylmethyl (CM) modification was first introduced by Grøtli et al. and was shown to give substantial stabilization of duplexes [16]. Milton et al. have shown that CM-modified oligonucleotides are also highly resistant to enzymatic degradation [17]. In addition, this scaffold can be further modified by different substituents on the amide nitrogen and several research groups investigated these possibilities [18,19,20,21,22,23,24]. The post-synthetic modification with 2′-*O*-(*N*-(aminoethyl)carbamoyl)methyl (2′-*O*-AECM) was investigated by Ozaki et al. [21]. Furthermore, our group has previously demonstrated the synthesis of 2′-*O*-AECM modified adenosine and that this modification provides a unique combination of resistance against enzymatic degradation and improved cell penetration properties for ONs [25]. In addition, the initial studies on 2′-*O*-AECM-modified ONs with all common 2’-*O*-AECM-modified ribonucleotides showed efficient splice-switching activity [26]. Moreover, as this modification showed promising properties for ON therapeutics [25,26,27], enabling its efficient synthesis, and upscaling is essential for moving oligonucleotide constructs forward towards the quantities required for animal experiments.

Herein, we report on our recent efforts to develop convenient procedures to prepare 5-methyluridine (AECM-MeU) and 5-methylcytidine (AECM-MeC) (Figure 1) monomers in larger (up to 100 g for some intermediates) scales. Furthermore, AECM-MeU and AECM-MeC are prepared as phosphoramidite derivatives that contain protecting groups (e.g., dimetoxytrityl and trifluoroacetyl on the amine) which enables them to be used as building blocks for automated solid-phase oligonucleotide synthesis.

5-Methylated AECM-modified uridine and cytidine phosphoramidites (Figure 1) have not been reported before, but the synthesis of the corresponding non-methylated AECM-modified cytidine and uridine nucleosides was achieved in scales of milligrams and a few grams, respectively [26,28]. However, several steps used for the non-methylated derivatives were considerably less efficient with the 5-methyl nucleosides and gave more problems with purifications. These findings, together with a preferred use of less expensive reagent, led to the need for a complete revision of the syntheses of AECM-MeU and AECM-MeC that are also more suitable for significantly larger scales. Moreover, more efficient purification and/or work-up strategies were necessary to be able to adapt the syntheses for larger scales. Herein, the developed procedures that enabled synthetic routes with fewer chromatographic purifications required only three chromatographic purifications for the intermediates up to the AECM-MeU nucleoside preceding the phosphoramidite conversion (in eight steps of synthesis from MeU) and only one for the intermediates up to the AECM-MeC nucleoside preceding the phosphoramidite conversion (in eight steps of synthesis from MeC) was needed. This was achieved by finding suitable conditions for work-ups, forming crystalline intermediates or finding strategies to perform one-pot synthesis. However, the final phosphoramidites were chromatographed prior to oligonucleotide synthesis (AECM-MeC phosphoramidite twice) to achieve a purity needed to ensure high-quality ON.

## 2. Results and Discussion

### 2.1. Synthesis of AECM-MeU Phosphoramidite

The study started with the development of a synthetic route for AECM-MeU phosphoramidite **7** (Figure 1). All reactions, starting with the N^3^ protection of 3′,3′,5′-O-[(1,1,3,3-tetraisopropyl-1,3-disiloxanediyl)]-5-methyluridine, except the final phosphoramidite synthesis, were either screened for optimized conditions based on quality attributes or tested in up to 1 g scale for feasibility before upscaling the reactions further. All of the screening reactions for individual steps were performed in 0.25–1 g scale and were evaluated using approximate conversions (%) from UV traces in the HPLC chromatogram (without internal standard).

The synthesis of AECM-MeU started with preparation of 3′,5′-*O*-[(1,1,3,3-tetraisopropyl-1,3-disiloxanediyl)]-5-methyluridine (**1**) [29] in 195 g scale. Although the protection of N^3^ with the pivaloxymethyl (Pom) group using phase transfer catalysis (PTC) was previously described [30], the development of this step started with screening for optimized conditions for the PTC reaction (Table 1). The goal was to reduce the amounts of solvents (as more than 120 volumes in total were used in a published procedure which is not satisfactory for large-scale synthesis) as well as to decrease the amount of chloromethyl pivalate (Pom-Cl). 

Evaluation of N^3^ protection of the AECM-MeU monomer was performed using different base–solvent combinations and tetrabutylammonium bromide (TBABr) as a PTC catalyst (Table 1). Although tetrabutylammonium hydrogen sulfate (TBAHS) was used in a published procedure, TBABr showed satisfying results in some of our initial screens and therefore we decided to continue with this PTC catalyst. In addition, some pre-screen tests showed that an increased reaction temperature could be beneficial, therefore the main screen (Table 1) was carried out at 50 °C. As one of the main goals was to evaluate if smaller amounts of Pom-Cl are sufficient for the synthesis (in a previously described procedure [30] more than 9.9 equiv. of Pom-Cl were used), the screening reactions were carried out using only a slight excess (1.5 equiv.) of this reagent (Table 1).

The study revealed that the reaction proceeds most rapidly in DMF in the presence of K_2_CO_3_ (Table 1, entry 8). However, extended reaction times led to degradation which indicates that the reaction must be monitored carefully. Nonetheless, based on the screenings, final conditions for the synthesis of compound **2** were 2 equiv. of chloromethyl pivalate, 4 equiv. of K_2_CO_3_, 0.2 equiv. of TBABr in DMF. It was decided to increase the amount of the base in order to have a larger particle surface and 2 equiv. of Pom-Cl was used to make sure the reaction went to completion. Although HPLC analysis revealed that using these conditions, after 70–75% conversion, the amount of side products substantially increases, different bases and solvents did not seem to be more advantageous. The possibility to have a rather fast reaction as well as to decrease the amount of solvent and Pom-Cl was regarded as more attractive and the above-mentioned conditions were chosen for larger scale synthesis, which was performed in 50 g scale for 6 h. In addition, the reaction was sampled after 5.5 h to give a rather clean HPLC profile (Appendix A).

PTC conditions were also employed for 2’-OH alkylation to obtain compound **3** and the process development to find optimal conditions of this step was performed similarly as for the N^3^ alkylation (Table 2). The screen was carried out using different base–solvent–PTC catalyst combinations. The results showed that a PTC catalyst is crucial for the reaction to proceed efficiently (Table 2, entry 1 and 2) with a rather minor difference between the three PTC agents tried (Table 2). As TBABr was working well for N^3^ alkylation and additional screens confirmed its efficiency (Appendix A), this catalyst was chosen for the larger scale reaction. Moreover, additional screens were performed to determine suitable amounts of base and methyl 2-bromoacetate (Appendix A). Different bases (K_2_CO_3_ and K_3_PO_4_) were also compared and revealed that the alkylation reaction results in the least amount of impurities when it is performed in heptane or acetonitrile (MeCN) in the presence of K_2_CO_3_. To confirm these results, additional screenings (Appendix A) were performed, which showed that a reaction in heptane was cleaner compared to MeCN (Appendix A). However, upon a transfer of conditions to a larger scale, solubility became an issue (5 vol of heptane was used in test reactions (Table 2, entries 6 and 11)), which did not seem a problem at the time. However, larger amounts of compound **2** took longer times to dissolve in 5 vol of heptane and the alkylation product **3** can be poorly soluble in this solvent. To overcome this problem, we chose to perform the reaction in a DCM/heptane mixture (this test reaction was done prior to larger scale (Table 2, entry 14)). 

Finally, after the series of screens, 5 vol DCM/heptane (1:4 (*v*/*v*)), 4 equiv. of K_2_CO_3_ and 0.05 equiv. of TBABr were selected for the larger scale of MeU 2′-OH alkylation. The amount of K_2_CO_3_ was increased for the same reason as for N^3^ alkylation. Nonetheless, the scaling-up of a PTC reaction using solid base can sometimes become challenging. The grinding effect of the magnet in smaller scale PTC reactions is more pronounced and hence fresh particle surface is continuously exposed. We observed that the PTC alkylation on a larger scale did not go to completion after stirring at ambient temperature for 66 h compared to the small-scale reaction which was completed after 36 h (Table 2, entry 14). The additional added amounts of K_2_CO_3_, TBABr and methyl 2-bromocetate facilitated the reaction to go to completion. At this point, a one-pot three-step synthesis followed. The reaction sequence started with selective opening of the 5′position of compound **3** using TFA in THF:water (5:1, *v*/*v*).

Selective opening of the cyclic silyl protecting group is more advantageous for selective introduction of the 5′-*O*-DMT protection and subsequent steps in terms of yield and purity, as the retained silyl on the 3′-OH mitigated the risk of lactonization (2′-*O*-methoxyacetyl). Crude isolated product **4** was further treated with 4,4′-dimethoxytrityl chloride at ambient temperature. The reaction was then quenched with MeOH followed by addition of ethylenediamine (EDA) and the reaction mixture was then heated to 60 °C to complete aminolysis of the methyl ester and removal of the Pom protecting group from N^3^. After stirring overnight, TEAx3HF was added to remove the remaining silyl protecting group at the 3′-position. Following an aqueous work-up, crude **5** was acylated using ethyl trifluoroacetate and then chromatographed to afford compound **6** (59% from **3**, 29% overall yield from **1**, 84% average yield per step). Compound **6** was further phosphitylated at the 3′-position using standard conditions with 2-cyanoethyl *N*,*N*-diisopropylphosphoramidochloridite in the presence of DIPEA in THF to yield the AECM-MeU phosphoramidite **7**. 

### 2.2. Synthesis of AECM-MeC Phosphoramidite

The reported synthesis procedure [26,28] of the non-methylated AECM-C monomer was used as a starting point for the preparation of AECM-MeC **16** (Figure 2) but changes and substantial improvements were developed before the upscaling. All of the screening reactions for individual steps were also performed in 0.25–1 g scale and were evaluated using approximate conversions (%) from UV traces in the HPLC chromatogram (without internal standard).

The synthesis of AECM-MeC phosphoramidite **16** was carried out as shown in Figure 2. 3′,5′-*O*-[(1,1,3,3-Tetraisopropyl-1,3-disiloxanediyl)]-5-methylcytidine (**8**) was prepared starting from 100 g of 5-methylcytidine using a known procedure [31]. The procedure [26,28] for non-methylated AECM-C nucleoside continues with alkylation of the 2′-OH. However, for the synthesis of the AECM-MeC variant, the direct alkylation of the 2′-OH proved to be problematic due to poor solubility of compound **8** in DMF. However, it was found that conversion of the exocyclic amino group into an amidine, forming **9** [32], substantially increased the solubility, thus facilitating the 2′-*O*-alkylation reaction. 

Based on the experience from the 2′-*O*-alkylation presented above to the corresponding AECM-MeU nucleoside intermediate, we assumed that similar conditions could be applied for the preparation of AECM-MeC. Nonetheless, to assess which conditions were most suitable for this substrate, a screen of different bases and solvents was performed (Table 3).

The PTC catalyst for 2′-OH alkylation of the 5-methyl cytidine derivative remained the same. In addition, a couple of different solvents were screened (Table 3) and the results showed that both MeCN and heptane can be suitable solvents for 2′-OH alkylation of the protected MeC intermediate **9**. However, as the starting compound was only partially soluble in heptane and due to problems with the solubility of the final product, the conclusion was to try the same solvent system as for MeU (heptane/DCM mixture). Indeed, this gave a clean reaction and 94% conversion (Table 3, entry 8). 

In addition, although K_2_CO_3_ was performing well as a base, it turned out that the reaction was even faster and cleaner when K_3_PO_4_ was employed as base for 5-methyl cytidine 2′-alkylation (Table 3, entries 4–8).

As K_3_PO_4_ and DCM/heptane proved to give the most preferred outcome in terms of yield and impurity profile, these conditions were used to upscale the reaction to 50 g of **8**. Despite increasing the temperature to 40 °C, in order to shorten the overall reaction time, the reduced grinding effect of the magnet on a larger scale gave a reaction that still required approx. 50 h to go to completion as well as the addition of additional reagents. Crude alkylated compound **10** was reacted with ethylenediamine in methyl-THF to give intermediate **11** (including removal of the exocyclic amine protecting group) followed by washes using NH_4_Cl (aq.) to remove the excess of ethylenediamine. Then, crude **11** was subsequently treated with ethyl trifluoroacetate in methanol to afford compound **12.** The exocyclic amino group was acetylated to give **13**. However, this step, following a procedure for the corresponding AECM-C using acetic anhydride in pyridine [26,28], proved to require more attention as we observed around 20% of bis acetylated side product. Reduced added amounts of Ac_2_O did not improve the outcome as we noticed that the bis acetylated product forms in parallel with the desired mono acetylated compound. Due to this, we performed another screen to test different conditions for acetylation (different solvents, different amounts of Ac_2_O, different reaction times as well as the possibility to use TEA as a base (Appendix A)). However, this screen was unsuccessful in finding conditions for monoacetylation. Further evaluation of literature procedures revealed that in order to avoid bis acetylation, the reaction can be successfully carried out without the presence of any base [33]. Indeed, the slow addition of acetic anhydride without the presence of any base gave the monoacetylated derivative **13.** The crude product **13** was then treated with triethylamine trishydrofluoride in MeCN to give crystalline base-protected compound **14**. After stirring the reaction mixture overnight, tert-butyl methyl ether (TBME) was used as an anti-solvent to ensure a full precipitation of compound **14**, which was further purified by re-slurrying compound **14** in DCM. Compound **14** was then converted to 5′-*O*-4,4′-dimethoxytrityl derivative **15** using standard dimethoxytritylating conditions followed by flash column chromatography to obtain pure **15** (25% overall yield from **8**, 82% average yield per step). Tritylated compound **15** was then phosphitylated using standard conditions at the 3′-position to give the AECM-MeC phosphoramidite **16.**

## 3. Materials and Methods

### 3.1. General Information

Unless otherwise noted, all reagents and solvents used in chemical synthesis were of commercial grade.

***HPLC*** was performed on an UltiMate 3000 HPLC system (Thermo Fisher Scientific, Germering, Germany) with UV detection at 254 nm. Reversed-phase (RP) HPLC was performed on a XBridge^®^ Oligonucleotide BEH C18 (4.6 × 50 mm) column (Waters, Dublin, Ireland) with 1 mL/min flow rate.

***Chromatographic separations*** were performed on a CombiFlash^®^ Rf 200 by Teledyne ISCO system (Lincoln, NE, USA) using Biotage^®^ normal or reversed-phase (RP) silica columns (Uppsala, Sweden).

***Mass spectrometry analysis*** was performed with Waters Xevo QTof G2-XS (Dublin, Ireland). 

***NMR spectra*** were recorded using a Bruker AV 500 MHz (500.13 MHz in ^1^H, 125.76 MHz in ^13^C and 202.47 MHz in ^31^P, 470.56 MHz in ^19^F) spectrometer (Fällanden, Switzerland) using deuterated solvent signal as an internal standard. 1,2,4,5-Tetrachloro-3-nitrobenzene was used as an internal standard for quantitative NMRs (assay). Chemical shifts (*δ* scale) are reported in parts per million (ppm). Coupling constants (J values) are given in Hertz (Hz).

### 3.2. Synthesis of Compounds **2**–**7**

*N*^3^-Pivaloyloxymethyl-3′,5′-*O*-[(tetraisopropyldisiloxan-1,3-diyl)]-5-methyl-uridine (2).

Fifty grams (91 mmol) of 3′,5′-O-(1,1,3,3-tetraisopropyl-1,3-disiloxanediyl)-5-methyluri- dine (**1**) (91% assay by NMR, Appendix A) was dissolved in 250 mL dimethylformamide. K_2_CO_3_ (55.2 g, 399 mmol) was added to the solution followed by the addition of tetrabutylammonium bromide (6.44 g, 20.0 mmol). Then, chloromethyl pivalate (30.1 g, 200 mmol) was added dropwise to the reaction mixture under nitrogen atmosphere at ambient temperature. Then, the temperature of the reaction mixture was increased to 40 °C and left to stir for 6 h. K_2_CO_3_ was filtered off, and volatiles were removed under reduced pressure. The remaining crude material was re-dissolved in dichloromethane, organic phase was washed with saturated aq. NaHCO_3_ twice, dried over Na_2_SO_4_, filtered and evaporated under reduced pressure. The crude product was subjected to a column and chromatographed using 0 to 20% ethyl acetate in heptane. Fractions containing the product were concentrated and the isolated product was co-evaporated once with dichloromethane to give compound **2** (32.2 g, 52.4 mmol, 57.6%) as a white foam.

^1^H NMR (500 MHz, CDCl_3_): δ = 7.36 (d, *J* = 1.3 Hz, 1H), 6.00–5.91 (m, 2H), 5.73 (d, *J* = 1.1 Hz, 1H), 4.44 (dd, *J* = 8.4, 5.3 Hz, 1H), 4.20–4.14 (m, 2H), 4.08–3.97 (m, 2H), 2.88 (s, 1H), 1.94 (s, 3H), 1.19 (s, 9H), 1.13–0.96 (m, 28H). ^13^C NMR (202.47 MHz, CD_3_OD): 177.65, 162.62, 150.11, 134.93, 110.01, 91.68, 82.15, 77.73, 75.18, 69.60, 65.13, 60.71, 38.97, 27.17, 17.56, 17.51, 17.42, 17.39, 17.20, 17.13, 17.07, 17.00, 13.54, 13.34, 13.12, 12.85, 12.71 ppm. ES-MS calc. for C_28_H_50_N_2_O_9_Si_2_ [M+H]^+^ 615.3133, found 615.3139.

2′-*O*-(*O*-Methylcarboxymethyl)-*N**^3^*-pivaloyloxymethyl-3′,5′-*O*-[(1,1,3,3-tetraisopropyl-1,3-disiloxanediyl)methyl]-5-methyl-uridine (3).

Compound (**2**) (30.68 g, 49.9 mmol) was dissolved in a mixture of 123 mL heptane and 31 mL dichloromethane and the obtained solution was subsequently flushed with nitrogen. Methyl 2-bromoacetate (9.5 mL, 100 mmol) was slowly added to the stirred solution followed by the addition of tetrabutylammonium bromide (800 mg, 2.5 mmol) and K_2_CO_3_ (28 g, 200 mmol). The resulting suspension was stirred at ambient temperature for 66 h. As the reaction was not completed, additional methyl 2-bromoacetate (2.4 mL, 25 mmol), K_2_CO_3_ (13.8 g, 100 mmol) and tetrabutylammonium bromide (800 mg, 2.5 mmol) were added and the reaction was left to stir at ambient temperature for an additional 24 h. Then, 200 mL of water was added to the reaction mixture and phases were separated. Organic phase was washed with a 200 mL acetonitrile–water mix (1:1 *v*/*v*) which led to the formation of three phases. The phases containing product (middle and upper) were collected, dried over Na_2_SO_4_, filtered and evaporated to dryness in vacuo. The crude product was purified via flash chromatography using 0 to 100% ethyl acetate in heptane as eluent to afford compound **3** as a colorless oil (28.89 g, 42.06 mmol, 84.3%). ^1^H NMR (500 MHz, DMSO-*d*_6_) δ = 7.53 (s, 1H), 5.83–5.74 (m, 2H), 5.70 (s, 1H), 4.47 (d, *J* = 16.6 Hz, 1H), 4.36 (d, *J* = 16.5 Hz, 1H), 4.30 (dd, *J* = 9.2, 4.9 Hz, 1H), 4.22 (d, *J* = 5.0 Hz, 1H), 4.14 (dd, *J* = 13.5, 2.0 Hz, 1H), 4.03 –3.98 (m, 1H), 3.92 (dd, *J* = 13.5, 2.6 Hz, 1H), 3.63 (s, 3H), 1.82 (d, *J* = 1.2 Hz, 3H), 1.10 (s, 9H), 1.08–0.95 (m, 28H). ^13^C NMR (202.47 MHz, DMSO-*d*_6_): δ = 178.37, 170.07, 161.71, 149.73, 136.57, 110.17, 90.87, 83.49, 83.01, 70.23, 68.57, 66.10, 60.42, 52.34, 39.87, 33.07, 30.18, 27.41, 23.34, 18.04, 17.95, 17.77, 17.60, 17.50, 17.36, 14.72, 14.47, 14.26, 14.05, 13.92, 13.49 ppm. ES–MS calcd. for C_31_H_54_N_2_O_11_Si_2_ [M + H]^+^ 687.3344, found 687.3337.

5′-*O*-(4,4′-Dimethoxytrityl)-2′-*O*-[(*N*-(trifluoroacetamidoethyl)carbamoyl)methyl]-5-methyl-uridine (6).

Compound **3** (28.89 g, 42.06 mmol) was dissolved in 289 mL THF and 58 mL water. Trifluoroacetic acid (3.2 mL, 42 mmol) was slowly added to the resulting solution and the reaction mixture was stirred at ambient temperature for 18 h. Reaction was quenched with 144 mL pyridine and volatiles were evaporated under reduced pressure. The residue was co-evaporated with pyridine to remove residual water, re-dissolved (pyridine, 300 mL) and half of the solvent volume was evaporated. 4,4′-Dimethoxytrityl chloride (15.7 g, 46 mmol) was added to the resulting mixture and stirred for 2.5 h at ambient temperature. Then, the reaction was quenched with MeOH (150 mL) followed by the addition of ethylenediamine (56 mL, 840 mmol) to the resulting solution, and the temperature of the reaction was set to 60 °C and was left to stir for 16 h. Et_3_N(HF)_3_ (20.6 mL, 126 mmol) was added to the (60 °C) reaction mixture, stirred for a few minutes and then the reaction was cooled down to 25 °C and stirred for 2.5 h. Volatiles were evaporated in vacuo, the residue partitioned between saturated aq. NH_4_Cl and dichloromethane (250 mL each), phases separated, organic phase washed with saturated aq. NH_4_Cl and water (250 mL each), dried over Na_2_SO_4_, filtered and evaporated under reduced pressure. The crude obtained material was dissolved in dichloromethane (139 mL), and triethylamine (30.0 mL, 215 mmol) was added to the resulting solution followed by the addition of ethyl trifluoroacetate (25.0 mL, 210 mmol). Reaction was stirred for 2.5 h at ambient temperature, then washed with saturated aq. NaHCO_2_, phases were separated, organic phase was dried over Na_2_SO_4_, filtered and evaporated under reduced pressure. The crude product was subjected to a silica column and chromatographed using 0 to 20% methanol in dichloromethane containing 1% triethylamine. The obtained material had to be re-chromatographed and was subjected to a second silica column and chromatographed using EtOAc/MeOH (95:5 *v*/*v*) containing 1% triethylamine as eluent to give compound **6** as a white foam (18.78 g, 59% overall yield after five steps, 95% assay according to NMR). ^1^H NMR (500 MHz, CD_3_OD): *δ =* δ 7.74 (d, *J* = 1.3 Hz, 1H), 7.46–7.43 (m, 2H), 7.36–7.28 (m, 6H), 7.28–7.22 (m, 1H), 6.93–6.81 (m, 4H), 5.93 (d, *J* = 2.4 Hz, 1H), 4.51 (dd, *J* = 7.5, 5.2 Hz, 1H), 4.34–4.22 (m, 2H), 4.14 (dt, *J* = 7.6, 2.6 Hz, 1H), 4.10–4.07 (m, 1H), 3.78 (s, 6H), 3.54 (dd, *J* = 11.0, 2.1 Hz, 1H), 3.46–3.42 (m, 5H), 1.33 (d, *J* = 1.2 Hz, 3H). ^13^C NMR (202.47 MHz, CD_3_OD): δ = 172.29, 165.87, 159.85, 159.00, 151.76, 145.45, 136.47, 136.38, 136.21, 130.91, 128.96, 128.46, 127.62, 118.11, 117.39, 115.83, 113.74, 110.92, 89.16, 87.53, 84.31, 83.56, 70.35, 69.52, 62.49, 55.22, 39.85, 38.59, 11.66, 0.26. ^19^F NMR (470.56 MHz, CD_3_OD): *δ* = −77.38 ppm. ES-MS calc. for C_37_H_39_F_3_N_4_O_10_ [M+Na]^+^ 779.2516, found 779.2514.

3′-*O*-(*N**,N*-Diisopropylamino-(2-cyanoethoxy)phosphinyl)-5′-*O*-(4-methoxytrityl)-2′-*O*-[(*N*-(trifluoroacetamidoethyl)carbamoyl)methyl]-5-methyl-uridine (7).

Compound **6** (5.1 g, 6.7 mmol) was dissolved in 52 mL of anhydrous THF. The solution was cooled in an ice bath and *N*,*N*′-diisopropylethylamine (5.9 mL, 34 mmol) was added under nitrogen atmosphere followed by the addition of 2-cyanoethyl *N*,*N*-diisopropylphosphoramidochloridite (3.0 mL, 13 mmol) and the reaction mixture was stirred for another 3 h. The reaction mixture was quenched with 300 µL MeOH and filtered. The filtrate was concentrated under reduced pressure and the residue partitioned between DCM and saturated aq. NaHCO_3_. Phases were separated and aqueous phase was again extracted with DCM. Organic phases were pooled together and dried over Na_2_SO_4_. The crude product was purified via flash column chromatography using 0 to 15% acetonitrile in ethyl acetate containing 1% triethylamine as eluent. Fractions were evaporated and yielded a white foam which was re-slurried in heptane to give the final compound **7** (4.7 g, 4.91 mmol, 72.5%). ^31^P NMR (202.47 MHz, CDCl3): *δ* = 149.6, 148.4 ppm. ^1^H, ^13^C and ^19^F NMRs are presented in the Appendix A. ES–MS calc. for C_46_H_56_F_3_N_6_O_11_P[M]^−^ 955.3624, found 955.3601.

### 3.3. Synthesis of Compounds **9**–**16**

*N*^4^-Dimethylformamidono-3′,5′-*O*-[(1,1,3,3-tetraisopropyl-1,3-disiloxanediyl)]-5-methyl-cytidine (9).

In 600 mL pyridine, 50.1 g (100 mmol) of 3′,5′-*O*-[(1,1,3,3-tetraisopropyl-1,3-disiloxanediyl)] -5-methyl-cytidine (**8**) was dissolved, and 350 mL of the solvent was evaporated prior to addition of *N*, *N*-dimethylformamide dimethyl acetal (40 mL, 301.1 mmol). After stirring at ambient temperature for 2.5 h, the solvent was evaporated under reduced pressure, the obtained crude material was co-evaporated with toluene (×3) and dried in vacuum to give 60.6 g of crude product **9**. ^1^H and ^13^C NMR data are included in the Appendix A. ES-MS calc. for C_25_H_46_N_4_O_6_Si_2_ [M+H]^+^ 555.3034, found 555.3043.

*N*^4^-Dimethylformamidino-2′-*O*-(*O*-methylcarboxymethyl)-3′,5′-*O*-[(1,1,3,3-tetraisopropyl-1,3-disiloxanediyl)]-5-methyl-cytidine (10).

Crude compound **9** (58.6 g) was dissolved in a mixture of dichloromethane (120 mL) and heptane (470 mL). Methyl 2-bromoacetate (20 mL, 211.3 mmol) was slowly added to the stirred solution followed by the addition of tetrabutylammonium bromide (680 mg, 2.1 mmol) and K_3_PO_4_ (44.8 g, 210.5 mmol). The resulting suspension was stirred at 40 °C for 26 h and then additional methyl 2-bromoacetate (5.0 mL, 53 mmol), K_3_PO_4_ (44.8 g, 210.5 mmol) and TBABr (680 mg, 2.1 mmol) were added. The reaction was left to stir at 40 °C. After 18.5 h, one more portion of K_3_PO_4_ (44.8 g, 210.5 mmol) was added, stirred for an additional 8 h at 40 °C and then 500 mL of water was added to the reaction mixture. Phases were separated, organic phase washed with MeCN:water (1:1) and then water, dried over Na_2_SO_4_, filtered and evaporated. Obtained crude oil **10** (74.8 g) was used for the next step without any further purification. ES-MS calc. for C_28_H_50_N_4_O_8_Si_2_ [M+H]^+^ 627.3245, found 627.3232.

2′-*O*-(*N*-(Ethyl)carbamoyl)methyl-3′,5′-*O*-[(1,1,3,3-tetraisopropyl-1,3-disiloxanediyl])-5-methyl-cytidine (11).

Crude compound **10** (74.2 g) was dissolved in 750 mL Me-THF. One hundred and sixty milliliters (2.4 mol) of ethylenediamine was added to the reaction mixture. It was stirred at ambient temperature for 1 h and then washed twice with 370 mL sat. NH_4_Cl followed by water (2 × 370 mL). The organic phase was evaporated under reduced pressure. As the water washes were found to contain some product, it was extracted with dichloromethane (×2). The combined dichloromethane phases were dried over Na_2_SO_4_, filtered and evaporated. Dissolving the residues from the organic phases in DCM and re-evaporation gave compound **11** (59 g) as a pale yellow foam which was used in the next step without any further purification. ^1^H and ^13^C NMRs are presented in the the Appendix A. ES-MS calc. for C_26_H_49_N_5_O_7_Si_2_ [M+H]^+^ 600.3249, found 600.3255.

2′-*O*-(*N*-(Trifluoroacetamidoethyl)carbamoyl)methyl-3′,5′-*O*-[(1,1,3,3-tetraisopropyl-1,3-disiloxanediyl])-5-methyl-cytidine (12).

Crude compound **11** (59 g) was dissolved in 400 mL methanol. Ethyl trifluoroacetate (60 mL, 504.2 mmol) was slowly added to the reaction mixture followed by the addition of triethylamine (136 mL, 975.7 mmol). Reaction was stirred at ambient temperature for 1.5 h, volatiles were evaporated under reduced pressure, residue was co-evaporated with toluene (×3) and the obtained crude product **12** (74.2 g) was used for the next step without any further purification. ^1^H and ^13^C NMRs are presented in the Appendix A. ES-MS calc. for C_28_H_48_F_3_N_5_O_8_Si_2_ [M+H]^+^ 696.3072, found 696.3059.

*N*^4^-Acetyl-2′-*O*-[(N-(trifluoroacetamidoethyl)carbamoyl)methyl-3′,5′-*O*-[(1,1,3,3-tetraisopropyl-1,3-disiloxanediyl])-5-methyl-cytidine (13).

Crude compound **12** (74.2 g) was dissolved in 371 mL dichloromethane. Acetic anhydride (20.1 mL, 212.6 mmol) was slowly added to the reaction mixture and left to stir at ambient temperature for 23 h. Then, 100 mL of water was added to the reaction mixture and stirred for 7 min followed by the addition of 400 mL sat. NaHCO_3_. After stirring for a few minutes, the reaction mixture was transferred to a separating funnel, phases separated and organic phase washed (×2) with sat. aq. NaHCO_3_ and water. Organic phase was dried over Na_2_SO_4_, filtered and concentrated under reduced pressure. Obtained crude light orange foam **13** (66.6 g) was used for the next step without any further purification. ^1^H NMR is presented in the Appendix A. ES-MS calc. for C_30_H_50_F_3_N_5_O_9_Si_2_ [M+H]^+^ 738.3177, found 738.3159.

*N*^4^-Acetyl-2′-*O*-[(*N*-(trifluoroacetamidoethyl)carbamoyl)methyl]-5-methyl-cytidine (14).

Crude compound **13** (65.8 g) was dissolved in 329 mL of acetonitrile, triethylamine trishydrofluoride (28 mL, 170 mmol) was added and the reaction mixture was stirred at ambient temperature for 19 h. Precipitated product was filtered off, 400 mL of tert-butyl methyl ether was added to the filtrate as a co-solvent, left to stir over a weekend and filtered again. Products from both precipitations were pooled together, re-slurried in dichloromethane and left to stir overnight. The slurry was filtered to give 17 g of the product **14** (35% from **8**). ^1^H NMR (500 MHz, CD_3_OD): δ = 8.45 (s, 1H), 5.93 (s, 1H), 4.42–4.28 (m, 2H), 4.25 (s, 1H), 4.10 (d, *J* = 8.5 Hz, 1H), 4.03 (d, *J* = 12.5 Hz, 1H), 3.98 (d, *J* = 5.0 Hz, 1H), 3.83 (d, *J* = 12.5 Hz, 1H), 3.53–3.37 (m, 4H), 2.39 (s, 3H), 2.05 (s, 3H). ^13^C NMR (202.47 MHz, CD_3_OD): δ = 170.96, 156.75, 143.91, 117.97, 115.31, 93.87, 89.35, 84.31, 70.94, 69.49, 67.38, 62.64, 59.10, 40.40, 37.43, 26.59, 24.79, 11.17. ^19^F NMR (470.56 MHz, CD_3_OD): δ = −77.36 ppm. ES-MS calc. for C_18_H_24_F_3_N_5_O_8_ [M+H]^+^ 496.1655, found 496.1653.

*N*^4^-Acetyl-5′-*O*-(4,4′-dimethoxytrityl)-2′-*O*-[(*N*-(trifluoroacetamidoethyl)carbamoyl)methyl]-5-methyl-cytidine (15).

Compound **14** (17.4 g, 35.12 mmol)) was dried by co-evaporation with pyridine (×3), re-dissolved in pyridine (87 mL), cooled in an ice bath and 4,4′-dimethoxytrityl chloride (12.5 g, 36.9 mmol) was added as a solid to the reaction solution. After stirring the reaction mixture for 3 h in an ice bath, an additional portion of 4,4′-dimethoxytrityl chloride (0.59 g, 1.8 mmol) was added and left to stir at RT for 16 h. Reaction was quenched with 200 µL of MeOH, then the reaction mixture was diluted with sat. aq. NaHCO_3_ and extracted with ethyl acetate. The organic phase was washed with sat. aq. NaHCO_3_ (×2), water and brine, dried over NaSO_4_ and evaporated. Obtained crude pale foam was purified via flash chromatography (pre-equilibrated with 1% TEA in DCM) using a gradient from 0 to 25% MeOH in DCM in the presence of 1% TEA. Compound **15** was obtained as a pale yellow foam after evaporation of pure fractions. The foam was re-slurried in heptane to give compound **15** as a white powder (20.1 g, 72%, 92% NMR assay, residual solvents present). ^1^H NMR (500 MHz, CD_3_OD): δ = 8.19 (s, 1H), 7.45 (d, J = 7.1 Hz, 2H), 7.36–7.17 (m, 7H), 6.84 (d, J = 9.0 Hz, 4H), 5.89 (d, J = 0.9 Hz, 1H), 4.57 (dd, J = 9.1, 5.0 Hz, 1H), 4.44–4.34 (m, 2H), 4.22 (d, J = 11.4 Hz, 1H), 4.06 (d, J = 5.4 Hz, 1H), 3.74 (s, 6H), 3.60 (d, J = 11.2 Hz, 1H), 3.44 (s, 5H), 2.36 (s, 3H), 1.45 (s, 3H). ^13^C NMR (202.47 MHz, CD3OD): δ = 173.29, 172.98, 163.90, 160.35, 159.48, 159.19, 157.06, 142.91, 136.95, 136.76, 131.45, 129.53, 129.08, 128.22, 118.66, 116.38, 114.36, 107.82, 91.42, 88.03, 85.19, 83.78, 71.03, 62.33, 55.81, 13.73, 13.21. ^19^F NMR (470.56 MHz, CD_3_OD): δ = −77.18 ppm. ES-MS calc. for C_39_H_42_F_3_N_5_O_10_ [M+H]^+^ 798.2962, found 798.2963.

*N*^4^-Acetyl-3′-*O*-(*N*,*N*-diisopropylamino-(2-cyanoethoxy)phosphinyl)-5′-*O*-(4-methoxytrityl)-2′-*O*-[(*N*-(trifluoroacetamidoethyl)carbamoyl)methyl]methyl-cytidine (16).

Compound **15** (5.0 g, 6.5 mmol) was dissolved in acetonitrile (15 mL) and dichloromethane (25 mL). The solution was cooled in an ice bath and N, N-diisopropylethylamine (2.2 mL, 13 mmol) was added under nitrogen atmosphere. Then, 2-cyanoethyl *N*,*N*-diisopropylphosphoramidochloridite (2.8 mL, 13 mmol) was added dropwise and the reaction mixture was stirred on ice for 1 h, the ice bath was removed and reaction stirred at ambient temperature for an additional 1.5 h. The reaction was quenched with 260 µL MeOH, volatiles were evaporated, residue dissolved in ethyl acetate and washed with sat. aq. NaHCO_3_ (×2) and brine (×1). Organic phase was dried over Na_2_SO_4_, evaporated and purified on a silica gel column (1% triethylamine in DCM pre-equilibrated) using 0 to 20% MeOH in DCM with 1% triethylamine in both solvents. Obtained white foam was re-slurried in heptane to give a mix (5.6 g) of a final compound **16** (72%) and a hydrolysis product of 2-cyanoethyl *N*,*N*-diisopropylphosphoramidochloridite (28%) (Appendix A) as a white powder (estimated final yield of the product **16** from ^31^P NMR: 4.0 g, 4 mmol, 62%). The impurity was successfully removed on an RP column using 25 to 100% MeCN in H_2_O (with 1% TEA) (Appendix A). ^31^P NMR (202.47 MHz, CD_3_CN): *δ* = 151.05, 148.4 ppm. ^1^H, ^13^C and ^19^F NMRs are presented in the Appendix A. ES-MS calc. for C_48_H_59_F_3_N_7_O_11_P [M]^−^ 996.3890, found 996.3871.

## 4. Conclusions

This study reports on procedures to prepare AECM-modified 5-methyluridine and 5-methylcytidine monomers in multigram scales aimed for the use in automated solid-phase oligonucleotide synthesis. The developed synthetic route for AECM-MeU monomer requires few purification steps. 2′-*O*-Alkylation of compound **2** was substantially improved by using PTC conditions which enable the possibility to use a low-cost and more readily available reagent. Additionally, reduced volumes of solvents as well as amounts of reagents for syntheses of compounds **2** and **6** were achieved compared to reported syntheses for related compounds. Moreover, selective opening of 5’-position of compound **3** unlocks the possibility to do three steps in one pot, and four steps in total without chromatographic purification to give 59% yield of compound **6** over five steps. The synthesis of the AECM-MeC monomer is clearly more optimally developed that for the preparation of AECM-C [26,28]. Using other work-up strategies, introduction of an amidine protecting group (compound **9**) and the possibility to crystallize compound **14** allowed us to avoid chromatography throughout the synthesis until the last phosphoramidite forming step. PTC conditions were also used to alkylate compound **9** at the 2′-OH position which reduced the cost of the synthesis by exchanging the P1-t-Bu-tris(tetramethylene) phosphazene base (BTTP) with the much lower cost reagent K_2_CO_3_. Moreover, syntheses of 5-methylated AECM-modified uridine and cytidine monomers are reported for the first time, which now enable their incorporation into oligonucleotides. In addition, syntheses were performed at larger scales to reduce cost and time for making these monomers for further use in biological evaluation of AECM-containing oligonucleotides.

## Data Availability

The datasets used and/or analyzed during the current study are available from the corresponding author on reasonable request.

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
