# Peer review of "A Study on Synthesis and Upscaling of 2′-O-AECM-5-methyl Pyrimidine Phosphoramidites for Oligonucleotide Synthesis"

_molecules, 2021, doi:10.3390/molecules26226927_

Round 1
Reviewer 1 Report
Strömberg and coworkers reported a full synthetic protocol for the obtention, in a mutligram scale, of two pyrimidine phosphoramidites analogs, with potential application in oligonucleotide syntheses. They performed a large screening of the reaction conditions, including the upscaling. Moreover, they tried to avoid chromatographic purifications, decreasing the amounts of waste.
I recommend the article to be published in Molecules after addressing the following comments and items.
When reading the introduction, I missed the importance of the use of phosphoramidites of this type in oligonucleotide syntheses. The authors should stress the importance of the groups attached to the final molecules, i.e. phosphoramidites, DMT group, trifluoroacetamide… Moreover, some important references are missed, in which similar compounds are synthesized (ACIE, 1998, 37, 1288; Eur. J. Org. Chem. 2006, 3152, between others).
In the lines 70-71, authors claim that they only performed 3 and 1 chromatographic purifications. If one checks the Materials Section, you can count 4 chromatographic columns plus a re-purification for the synthesis of compound 7; and 2 purifications for compound 16 (including the RP column). This issue should be revised.
In Scheme 1 and 2, I suggest the inclusion of the final reaction conditions (time, temperature, equivalents of reagents) and the scale of the reaction (grams). In the discussion, the authors explained the screening of conditions, and sometimes, the best conditions of the table did not correspond with the conditions carried out in the large scale reaction (due to upscaling issues).
In the Supplemental Information, NMR spectra are missed, including the ones for the key compounds 7 and 13. The following spectra must be included:
1H, 19F and 13C of 7. Highly important, since the authors are claiming the synthesis of this main compound in a pure form.
13C of 9
1H and 13C of 11,12 and 13
1H, 19F and 13C of 16. Highly important, since the authors are claiming the synthesis of this main compound in a pure form.
Author Response
Dear Reviewer,
We greatly appreciate the comments and have revised the manuscript accordingly as specified below point by point.
I recommend the article to be published in Molecules after addressing the following comments and items.
When reading the introduction, I missed the importance of the use of phosphoramidites of this type in oligonucleotide syntheses. The authors should stress the importance of the groups attached to the final molecules, i.e. phosphoramidites, DMT group, trifluoroacetamide…
Moreover, some important references are missed, in which similar compounds are synthesized (ACIE, 1998, 37, 1288; Eur. J. Org. Chem. 2006, 3152, between others).
In the lines 70-71, authors claim that they only performed 3 and 1 chromatographic purifications. If one checks the Materials Section, you can count 4 chromatographic columns plus a re-purification for the synthesis of compound 7; and 2 purifications for compound 16 (including the RP column). This issue should be revised.
- We have now clarified this by explaining that more specifically that we meant until the last step of amidite conversion (of course it should then also be 8 steps of synthesis). Lines 74-79.
In Scheme 1 and 2, I suggest the inclusion of the final reaction conditions (time, temperature, equivalents of reagents) and the scale of the reaction (grams). In the discussion, the authors explained the screening of conditions, and sometimes, the best conditions of the table did not correspond with the conditions carried out in the large scale reaction (due to upscaling issues).
- These details are now added to both Schemes
In the Supplemental Information, NMR spectra are missed, including the ones for the key compounds 7 and 13. The following spectra must be included:
1H, 19F and 13C of 7. Highly important, since the authors are claiming the synthesis of this main compound in a pure form.
- NMR data for 7 has now been added
13C of 9
NMR adta is now included in the supplementary (an cited on Lines 375-376)
1H and 13C of 11,12 and 13
Reviewer 2 Report
Since oligonucleotides have become compounds of central interest in medicinal chemistry in recent years, their large-scale synthesis gained great attention. The authors reported in this paper an efficient and scaled synthesis of two new 5-methyluridine and 5-methylcytidine phosphoramidite derivatives. The methyl group at 5-C-position of the base was going to be beneficial with respect to stability and biological responses. Another common moiety in these nucleoside derivatives is the [2-(trifluoroacetylamino)ethylcarbamoyl]methyl group attached to the 2’-O-position. The authors have already prepared structural related nucleosides, like adenoside derivatives (references 23 and 25), and taking this as a starting point, they went to the synthesis of these two compounds, in multigram scales, with the aim of minimize the purification steps (avoiding column chromatography, if possible), reducing the amount of solvents, and using available and relatively cheap reagents and materials. The ultimate goal is the use of these compounds in automated solid-phase oligonucleotide synthesis. Most of the effort was devoted to optimizing reaction conditions for functional group protection-deprotection, and especially for 2’-O-alkylation.
It is worth mentioning that fully experimental details are reported in the “Materials and Methods” section, along with spectroscopic data. Copies of 1H-, 13C-, 31P- and 19F-NMR spectra for some compounds are also provided in the “Supporting Information”.
I found this study of interest for people working in the synthesis of oligonucleosides and in medicinal chemistry and, therefore, I recommend the publication of this paper after addressing the following comments and some minor remarks that should be clarified.1. The measured value in HRMS should agree with the calculated value probably to at least the third decimal place. However, it is not the case in compounds 6 (cal 779.2516, found 779.2491: not bad), 7 (cal 955.3642, found 955.3950: to large) and 16 [cal 996.3890, found 996.4222 (996.422 in SI): to large]. 2. Why are not provided 1H- and 13C-NMR data for compounds 7 and 16?
Author Response
Dear Reviewer,
We greatly appreciate the comments and have revised the manuscript accordingly as specified below point by point.
Reviewer 2
I found this study of interest for people working in the synthesis of oligonucleosides and in medicinal chemistry and, therefore, I recommend the publication of this paper after addressing the following comments and some minor remarks that should be clarified.
- The measured value in HRMS should agree with the calculated value probably to at least the third decimal place. However, it is not the case in compounds 6 (cal 779.2516, found 779.2491: not bad), 7 (cal 955.3642, found 955.3950: to large) and 16[cal 996.3890, found 996.4222 (996.422 in SI): to large].
These now correspond to: for 6 calc: 779.2516, found 779.2514; for 7: calc 955.3642, found 995.3601; for 16: calc 996.3890, found 996.3871. The last two deviate on the third decimal place, but at these molecular weights it corresponds to ca 4 and 2 ppm respectively, which we think should be sufficiently accurate (a typical guideline is usually within 5 ppm).
- Why are not provided 1H- and 13C-NMR data for compounds 7 and 16?
- NMR data is now added for these compounds.